# Canadians Adults Fail Their Dietary Quality Examination Twice

**DOI:** 10.3390/nu15030637

**Published:** 2023-01-26

**Authors:** Gerard Ngueta, Caty Blanchette, Myrto Mondor, Jean-Claude Moubarac, Michel Lucas

**Affiliations:** 1Department of Community Health Sciences, Faculty of Medicine, Research Center of the CHU de Sherbrooke, Université de Sherbrooke, Sherbrooke, QC J1K 2R1, Canada; 2Research Center CHU de Québec, Université Laval, Quebec City, QC G1S 4L8, Canada; 3Department of Nutrition, Faculty of Medicine, Université de Montréal, Montréal, QC H3C 3J7, Canada; 4Department of Social and Preventive Medicine, Faculty of Medicine, Université Laval, Quebec City, QC G1V 0A6, Canada

**Keywords:** diet quality, alternative Healthy Eating Index (aHEI-2010), Canadian Community Health Survey

## Abstract

For many years, dietary quality among Canadians has been assessed using an index that gives criticized scores and does not allow for comparison with Americans. In Canadians aged ≥19 years, we aimed to (1) determine the dietary quality by using a more widely used evidence-based index that has shown associations with health outcomes, the alternative Healthy Eating Index (aHEI-2010); (2) assess changes in aHEI-2010 score and its components between 2004 and 2015; and (3) identify factors associated with aHEI-2010 score. We relied on the Canadian Community Health Survey 2004 (n = 35,107) and 2015 (n = 20,487). We used adjusted linear models with a time effect to compare the total aHEI-2010 score and its components. The overall aHEI-2010 score increased from 36.5 (95%CI: 36.2–36.8) in 2004 to 39.0 (95%CI: 38.5–39.4) in 2015 (*p* < 0.0001). Participants with less than a high school diploma showed the lowest score and no improvement from 2004 to 2015 (34.8 vs. 35.3, *p* = 0.4864). In each period, higher scores were noted among immigrants than non-immigrants (38.3 vs. 35.9 in 2004, *p* < 0.0001; 40.5 vs. 38.5 in 2015 *p* < 0.0001), and lower scores were observed in current smokers (33.4 vs. 37.1 in 2004, *p* < 0.0001; 34.5 vs. 39.9 in 2015, *p* < 0.0001). The use of the aHEI-2010 tool suggests a lower score among Canadians than the previous index, more comparable to the score among Americans.

## 1. Introduction

Dietary factors are among the highest risk factors of premature death and disability [1]. Inadequate diet quality is the main cause of premature death worldwide, and improvement in dietary quality has been associated with lowered premature death risk [2,3,4]. Prevention strategies cannot be effective without population-based measures to assess the most relevant nutritional criteria linked to early death. 

Assessing population trends in diet quality is essential because it provides an evidence base and simple strategic directions for improving public health policies. However, the definitions and tools for determining diet quality generate much questioning. Some tools can sometimes give imprecise results, suggesting that the diet quality of a population is fairly decent. In 2012, the *Dietary Patterns Methods Project* was launched in the United States to strengthen the evidence on the best different dietary quality indices [5]. A wide range of indices were considered, and three of them were outstanding in their superiority: the alternative Healthy Eating Index 2010 (aHEI-2010), the alternate Mediterranean Diet (aMED) and the Dietary Approaches to Stop Hypertension (DASH). Furthermore, the 2015 Dietary Guidelines for Americans [6] recommend the use of these quality indices to define and assess dietary patterns because they are practical and allow for tangible actions for the public. A comprehensive analysis of dietary quality required a multidimensional index commonly used in the literature. Such tools are the combination of most important nutritional factors in relation to population health.

For many years, dietary quality among Canadians has been assessed using the Canadian-Healthy Eating Index (C-HEI) [7], which has recently been criticized [8]. According to Lucas and Willett [8], the C-HEI-2005 aims to reflect dietary recommendations of the Canadian Food Guide 2007 (CFG), which does not incorporate the current best available evidence on diet and health, thus does not provide an appropriate reference to establish a valid diet-quality index. In addition, its use in Canada makes comparisons between the Canadian population and other countries impossible.

Based on a nationally representative sample of 29,124 U.S. adults (20 to 85 years), Wang et al. reported that aHEI-2010 scores showed steady improvement across an 11-year period (1999–2010) and was positively associated with both family income and education level [9]. Data based on recommended indices (e.g., aHEI-2010) in the Canadian population are scarce. Further analyses using the aHEI-2010 to assess diet quality would be useful to gain a better understanding of diet quality scores among Canadians and would provide evidence-based benchmarks for comparison with U.S. data. In this study, we used a large Canadian population survey with objectives to (1) determine the dietary quality of Canadian adults by using the aHEI-2010, (2) assess changes in aHEI-2010 score and its components between 2004 and 2015, and (3) identify factors associated with aHEI-2010 score.

## 2. Materials and Methods

### 2.1. Study Design and Population

The Canadian Community Health Survey (CCHS) is an annual population-level survey conducted by Statistic Canada since 2000 [10]. The CCHS is a complex survey with two cross-sectional components conducted in 2004 and 2015, as described elsewhere [11,12]. As part of the survey, the CCHS-Nutrition (CCHS-N) conducted in 2004 was the first national survey of the eating habits of Canadians, which has been conducted since the early 1970s. Nutrition was the topic of the CCHS in 2004 (n = 35,107) [11] and 2015 (n = 20,487) [12] and provided a significant source of national and provincial-level data, including food and nutrient intakes. All foods reported by participants as having been consumed in 24 h were compiled in grams and rated in terms of the standard serving size. We excluded pregnant and breastfeeding women and those who did not eat on the day of interview. 

### 2.2. Dietary Assessment—Alternative Healthy Eating Index 2010 (aHEI-2010)

The aHEI-2010 is based on foods and nutrients and contains 11 components with scores from 0 (worst) to 10 (optimal) [13]. Because *trans*-fatty acid has been forbidden in North America since 2015 and its reduction contributed more than half of the improvement in the overall aHEI-2010 in the U.S. population [9], it was then removed from our total score. The remaining 10 items and the scoring method of the aHEI-2010 (maximum score of 100) are described in Table 1. In the original version of the aHEI-2010, the sodium score is counted using the decile distribution in the population under study. Because population intakes do not always represent the thresholds and guidelines issued by health authorities, this can lead to misleading score values. In the current study, the sodium component has been modified and is based on guidelines from the Institute of Medicine [14] and is expressed per calories consumed per day [7].

### 2.3. Statistical Analyses

We used adjusted linear models with a time effect to compare dietary intake and total aHEI-2010 score and its components between 2004 and 2015. The effect of different participants’ characteristics (including province of residence) over time on total aHEI-2010 score was estimated, separately for each characteristic, by including effects for characteristic, time, and characteristic-by-time interaction, plus adjustment variables. All models were adjusted for sex, age, income, education, food insecurity, marital status, region status, immigrant status, current smoking status, body mass index (BMI), household size, and energy intake (continuous). All analyses were weighted and performed in SAS 9.4 using the surveyreg procedure with bootstrap weights and balanced repeated replication variance to account for the complex survey design. 

## 3. Results

A total of 19,610 adults (in 2004) and 13,439 (in 2015) were included in the adjusted linear models. The adjusted aHEI-2010 scores from a representative sample of Canadians aged ≥ 19 years are shown in Table 2. The overall aHEI-2010 score increased from 36.5 (95%CI: 36.2–36.8) in 2004 to 39.0 (95%CI: 38.5–39.4) in 2015 (*p* < 0.0001), but some component scores did not change (fruits, whole grains, and red/processed meat), and even deteriorated (vegetables and alcohol). However, an improvement in aHEI-2010 component scores was noted for SSB and fruit juice, nuts and legumes, long-chain n-3, PUFA, and sodium.

Table 3 shows statistically significant interactions between time and age categories (interaction *p* = 0.0019), household income (interaction *p* = 0.0450), education level (interaction *p* = 0.0087), and smoking status (interaction *p* = 0.0214). Table 3 also shows statistically significant differences in 2015 aHEI-2010 scores in most participants’ characteristics, with marital status and region status being the exceptions. Participants with the lowest education levels (< high school diploma) had the lowest diet quality and were the only ones who did not experience a diet quality improvement over time (34.8 in 2004 vs. 35.3 in 2015, *p* = 0.4864), whereas there was an improvement in the score for all other participants (interaction *p* = 0.0087), with those with ≥ Bachelor’s Degree having the highest diet quality. All household income quintiles showed an increase in the adjusted aHEI-2010 score between 2004 and 2015, but higher quintiles showed a larger increase (interaction *p* = 0.0450). Some other specific subgroups showed no change in the adjusted aHEI-2010 score from 2004 to 2015: adults aged ≥71 years, those with food insecurity, and current smokers. For age and current smoking status, this led to a statistically significant interaction (interaction *p* = 0.0019 and 0.0214, respectively), because an increase was observed in the other age groups and participants who smoked. In each period, higher scores were noted among immigrants compared to non-immigrants (38.3 vs. 35.9 in 2004, *p* < 0.0001; 40.5 vs. 38.5 in 2015, *p* < 0.0001), and the lowest scores were noted in current smokers vs. participants not currently smoking (33.4 vs. 37.1 in 2004, *p* < 0.0001; 34.5 vs. 39.9 in 2015, *p* < 0.0001). 

According to analysis by province (Table 4), we found that energy intake was stable or lower over time in most of the provinces. We also found that the aHEI-2010 scores were lower in the other Canadian provinces than in British Columbia, which had the maximum score value of all provinces (38.1 for 2004 and 42.0 for 2015). The highest improvement in aHEI-2010 scores from 2004 to 2015 was observed in British Columbia (increase of 4.0) and Alberta (increase of 3.7), and the lowest improvement was in Manitoba (increase of 1.5), Quebec (increase of 1.6), New Brunswick (increase of 1.7), and Saskatchewan (increase of 1.7). In the last survey (2015), only 21.8% of participants had a total aHEI-2010 score ≥50/100 (Table 5). In addition, more than half of the 2015 participants had an aHEI-2010 item score <5/10 for vegetables, fruits, whole grains, nuts and legumes, long-chain n-3, and alcohol.

## 4. Discussion

Despite a slight improvement (+2.5) in diet quality from 2004 to 2015 among Canadians aged ≥ 19 years, our overall findings suggest that diet quality remains poor. We also found that energy intake has been stable or lower over time in most of the provinces. More than three quarters of the participants had an aHEI-2010 score below 50/100 in 2015. The lowest aHEI-2010 scores (reflecting poor diet quality) in 2015 were observed in young adults in those with the lowest education levels and in smokers. Our observations are in line with the findings of Wang et al. who reported, among a nationally representative sample of 29,124 U.S. adults (20 to 85 years), that aHEI-2010 scores showed a steady improvement across an 11-year period ( +2.9 from 1999–2000 to 2009–2010) and was positively associated with both family income and education level [9]. The diet quality score in 2003–2004 was 34.9/100 (95%CI: 33.9–35.9) and 37.1/100 (95%CI: 36.6–37.7) in their last measurement in 2009–2010. These results are of a similar magnitude to those observed in our study in Canadian adults (36.5/100 in 2004 to 39.0/100 in 2015).

However, using the same CCHS-2004 data with C-HEI gives a completely different picture and indicates a much higher diet quality score. The average diet quality in Canadians, based on C-HEI-2005, instead reported a score of 58.8/100 [7]. This result is 1.6 times higher than our results with the aHEI-2010 and is 1.7 times higher than the American results. More recently, a similar level of diet quality score of 54.4/100 has been reported by using the C-HEI-2005 in a sample of 1147 French-speaking adults from the province of Quebec in 2015–2017 [15]. Using the aHEI-2010, in the present study, we noted a diet quality score of 37.7/100 (95%CI: 36.7–38.6) among adults in the province of Quebec in 2015. The score derived from the C-HEI-2005, with a value above the middle score of 50, gives an estimation of diet quality that is probably overly optimistic for the following reasons. According to Lucas and Willett [8], the appropriateness of assigning the maximum score (10 points) for two to four servings of milk products and one to three servings of meat and alternatives is highly questionable. Moreover, evidence on health outcomes does not support including potatoes as vegetables, juice as fruit, and legumes, nuts and fish in the same category as red meat. Further, no negative points are given for red meat and/or processed meat and for juice and sugary drinks, and there is no basis for giving a maximum score for a low-fat diet. Thus, some criteria of the C-HEI-2005 are irrelevant to the risk of early death. These criteria are based on the former CFG, which was highly criticized for its narrow approach to the food industry products [16,17,18].

In 2019, Health Canada released its new Canada’s Food Guide (CFG-2019) [19], which was a hybrid of the 2011 Harvard Healthy Plate [20] and the 2015 Brazilian Food Guide [21]. Recently, a new index, the Healthy Eating Food Index (HEFI-2019), has been developed to measure the adequacy of dietary intakes with the most recent Canadian recommendations (CFG-2019) [22]. Using the CCHS-2015 data among Canadians aged ≥19 years, the mean total score to HEFI-2019 was 43.3/80 for males and 46.0/80 for females [23]. The ratio of intakes (proportion) was used as an approach instead of quantity. Despite the great improvement over the former Canadian index (C-HEI-2005) [7] and the relevance and scientific evidence of the criteria used for the HEFI-2019, most of them are general recommendations (e.g., max. score if total vegetables and fruits/total foods ≥0.5). More importantly, these general recommendations are based on total food calculation in the denominator, making the determination very difficult, if not impossible, for clinicians. Furthermore, this index has never been validated and associated with health outcomes in a longitudinal cohort study or clinical intervention.

Over this 11-year period, some components worsened (vegetables and alcohol), and others remained unchanged (fruits, whole grains, and red and processed meat). As already stated, the lowest aHEI-2010 scores (reflecting very poor diet) were observed in young adults, in those with the lowest education levels and in smokers. This is consistent with previous findings [9,24,25,26] and may be partly due to lower nutrition knowledge [27], food prices [28], and food agency [29]. Previous studies also suggested that smokers consume fewer fruits and vegetables than non-smokers do [30]. It is noted in the literature that diet quality varied among population subgroups, especially for several indicators of socioeconomic status (SES) such as household income, education, and food insecurity. Nutrition inequities were also noted among the U.S. adults [9] and U.S. adults with diabetes [31], in which SES was strongly associated with a lower dietary quality score. We also observed similar results: diet quality improved over time in the higher-income and education groups, but in the lower income groups, no significant temporal improvement was observed. Results from Tarasuk et al. [32], using data from the 2004 CCHS, suggest that nutritional quality is, in part, a function of social position. For the past decade, price and taste have been the most important determinants of food purchasing and consumption decisions [33]. The significant increase in food prices following the various waves of COVID-19 and the war in Ukraine may worsen this disparity. 

Significant improvements for SSB and fruit juice, nuts and legumes, long-chain n-3, PUFA, and sodium contributed to the overall improvement in the aHEI-2010 score. However, a significant proportion (45.8–81.7%) of the Canadian population still scored <5/10 on these items in 2015. In addition, intakes of fruits, whole grains, and red/processed meat did not improve between 2004 and 2015, and intakes of vegetables deteriorated. Thus, in 2015, 67.9%, 74.1%, and 80.8% of the population had a score <5/10 for their vegetables, fruits, and whole-grain intakes, respectively. The overall picture of diet quality in Canada indicates that more efforts and innovative strategies are needed to improve the situation. Unfortunately, instead of addressing the powerful commercial and environmental determinants of eating habits [34], the focus is often on information strategies and approaches that place the responsibility on the ability of individuals. As mentioned by Mozaffarian et al. [35], “*Not surprisingly, this strategy has fallen short, as demonstrated by the increasing rates of obesity, diabetes, and other diet-related illness*”. Instead of focusing on individual nutrients or calories, food and dietary patterns represent more attainable, evidence-based goals. Beyond diet quality indexes, we need to integrate approaches that normalize the vision that healthy eating must be shared as a vital mission for oneself and others, but for that, we also need to promote upstream cooking at home as a privilege for the senses and health of all. A good start would be to consider food agency as empowerment to act and cook [36]. An interesting observation from our analyses is the fact that a higher diet quality score was noted among immigrants, which might indicate that they likely have a different food culture and cooking habits that help including more plant sources and less ultra-processed products.

The alcohol item in diet quality scales is still contentious. However, the most commonly used indexes in the literature, the aHEI-2010 and Mediterranean diet, include a criterion for alcohol. In moderation, alcohol can be consumed as part of an overall healthy diet and has been associated with a lower risk of various health problems (all-cause and CVD mortality, coronary heart disease, diabetes, dementia, etc.) [13]. However, heavy drinking produced the opposite effect and contributed to an increase in the Global Burden of Disease (cancers, accidents, etc.) [37,38]. Another side effect of alcohol is that it can contribute significantly to higher energy intakes and thus promote adverse effects on weight and waist circumference. Using 2017–2018 NHANES data, Warner et al. [39] observed that despite a slightly higher caloric intake from alcohol in moderate drinkers, non-statistically significant differences in BMI and WC were observed compared to non-drinkers and never drinkers. Similar observations were noted with the Mediterranean diet assessment tool and obesity indexes among high-risk subjects in the PREDIMED Trial [40]. A lower average BMI, waist circumference and waist-to-height ratio were noted among those who drink wine (≥7 glasses/week). 

Our use of health-authority-determined thresholds for sodium instead of a population distribution of intakes is an improvement over the original aHEI-2010. Population intakes do not always represent the thresholds and guidelines issued by health authorities [14], and this can lead to misleading scoring. Our calculation was based on these thresholds and expressed per calories consumed per day [7], and it is less likely to lead to the overestimation of the score for sodium than the original version.

Our study has limitations that should be considered. First, 24 h dietary recall was used a methodology to assess dietary intake among Canadians. The results therefore represent the behavior of individuals on a single day and not their usual behavior. However, Statistics Canada surveys, such as the NHANES, use this methodology to capture usual intakes at the population level. This information about diet, which is representative of our populations, remains the primary source on nutritional status. Second, we only had access to two points in time (2004 and 2015), which impeded our ability to identify trends. We had to use comparisons between the two times. Third, our results had to be interpreted in the context of the limitations and strengths of any cross-sectional study design. Therefore, we could not ascertain any causal relationship. Four, the cross-sectional nature of the study design enabled us to investigate longitudinal data for individuals. Our study also has strengths, including a representative sample of the Canadian population (with weights) and a large sample size. However, a significant advantage of using the aHEI-2010 is that it is still relevant and represents the latest evidence in the nutritional domain, despite its first publication a decade ago [13]. Moreover, the aHEI-2010 items are still aligned with the 2021 Dietary Guidance and Scientific statement from the American Heart Association [4].

## 5. Conclusions

Dietary quality indices that include several validated health-related items are important tools for monitoring population health. To achieve this, we must be able to distinguish in their ratings between foods that have health benefits and those that do not. This does not seem to be the case for the C-HEI-2005 and might explain why it gives an overestimated score. For clinicians and policy makers, validated tools that adequately evaluate diet quality are useful in guiding us toward the most appropriate approaches to improve the dietary scores of individuals or groups of individuals in our populations. Unfortunately, these multidimensional dietary tools are not widely used to monitor the status (deterioration or improvement) of overall nutrition. Nevertheless, low dietary quality is one of the most important contributors to premature death in our societies. Given escalating food prices and socioeconomic disparities, a lack of action on environmental and behavioral determinants, and food agency, poor diet quality in the Canadian population is likely to remain a concern for many years to come.

In conclusion, our results show that Canadians failed their diet quality test twice, both in 2004 and 2015. The use of the aHEI-2010 suggests a lower diet quality score among Canadians than that obtained with the C-HEI-2005, more comparable to the score among Americans. Behind the slight increase in the aHEI-2010 score in 2015 compared to 2004, it appears that not all individuals benefited from the same improvement in their food quality score, indicating disparities. In order to inform targeted policies and interventions, further analyses are required to address the disparity observed across provinces of Canada to investigate the diet quality of immigrants and the relationship between diet quality, income, and educational level. As a whole, significant efforts should be devoted to improving diet quality in Canadian adults.

## Figures and Tables

**Table 1 nutrients-15-00637-t001:** Components and criteria for scoring the alternate Healthy Eating Index 2010 (aHEI- 2010).

Components	aHEI-2010 Scoring	
	Minimal (0)	Maximal (10)
Vegetables, serving/day	0	≥5
Fruits, serving/day	0	≥4
Whole grains, g/day		
Women	0	≥75
Men	0	≥90
Nuts and legumes, serving/day	0	≥1
Long-chain omega-3 fatty acids (EPA + DHA) ^a^, mg/day	0	≥250
PUFA ^b^, % total energy	≤2	≥10
SSB ^c^ and fruit juice, serving/day	≥1	0
Red/processed meat, serving/day	≥1.5	0
Sodium, mg/1000 kcal	≤700: score = 10	
	700 to 1100: score = 10 to 8	
	1100 to 2000: score = 8 to 0	
	≥2000: score = 0	
Alcohol, drinks/day	Women	Men
	0: score = 2.5	0: score = 2.5
	(0–0.5): score = 5	(0–0.5): score = 5
	[0.5–1.5]: score = 10	[0.5–2.0]: score = 10
	(1.5–2.0): score = 5	(2.0–2.5): score = 5
	[2.0–2.5): score = 2.5	[2.5–3.5): score = 2.5
	≥ 2.5: score = 0	≥3.5: score = 0
**Total score**	**0**	**100**

**^a^** EPA + DHA stands for eicosapentaenoic acid and docosahexaenoic acid; **^b^** PUFA stands for polyunsaturated fatty acid (excluding omega-3 fatty acids); **^c^** SSB stands for sugar-sweetened beverages.

**Table 2 nutrients-15-00637-t002:** Adjusted **^a^** mean dietary intake and alternate Healthy Eating Index 2010 (aHEI-2010) scores and 95% confidence intervals (95% CI) in adults from the Canadian Community Health Survey (CCHS)-Nutrition in 2004 and 2015.

	2004 Mean (95% CI)	2015 Mean (95% CI)	*p* for Difference
**Dietary Intake**			
Total energy intake, Kcal/d	2076 (2049–2102)	1896 (1869–1924)	<0.0001
Vegetables, servings/d	2.5 (2.3–2.6)	2.1 (1.9–2.2)	<0.0001
Fruits, servings/d	1.3 (1.2–1.4)	1.3 (1.1–1.4)	0.1133
Whole grains, g/d	19.4 (18.5–20.4)	21.9 (20.5–23.4)	0.0057
Nuts and legumes, servings/d	0.33 (0.26–0.40)	0.44 (0.37–0.52)	<0.0001
SSB ^b^ and fruit juice, servings/d	1.4 (1.3–1.5)	1.1 (1.0–1.3)	<0.0001
Long-chain n-3 (EPA + DHA) ^c^, mg/d	151 (136–167)	159 (138–180)	0.5838
PUFA ^d^, % of energy	5.6 (5.4–5.8)	6.9 (6.7–7.2)	<0.0001
Red/processed meat, servings/d	0.94 (0.86–1.02)	0.92 (0.85–1.00)	0.5458
Sodium, mg/d	2897 (2859–2934)	2802 (2766–2839)	0.0005
mg/1000 kcal	1545 (1524–1567)	1486 (1466–1506)	<0.0001
Alcohol, drink/d	0.58 (0.42–0.75)	0.66 (0.52–0.81)	0.0310
**aHEI-2010 score**			
Vegetables (/10)	4.3 (4.1–4.5)	3.8 (3.6–4.0)	<0.0001
Fruits (/10)	2.9 (2.7–3.0)	2.8 (2.6–3.0)	0.5061
Whole grains (/10)	2.1 (1.9–2.2)	2.2 (2.0–2.4)	0.0609
Nuts and legumes (/10)	2.0 (1.8–2.2)	2.6 (2.4–2.8)	<0.0001
SSB ^b^ and fruit juice (/10)	4.6 (4.3–4.9)	5.2 (4.9–5.5)	<0.0001
Long-chain n-3 (EPA + DHA)^c^ (/10)	2.4 (2.2–2.6)	2.6 (2.4–2.8)	0.0329
PUFA ^d^ (/10)	4.3 (4.1–4.5)	5.6 (5.4–5.8)	<0.0001
Red/processed meat (/10)	5.7 (5.5–5.9)	5.9 (5.6–6.1)	0.0565
Sodium (/10)	4.8 (4.6–5.0)	5.0 (4.8–5.2)	0.0092
Alcohol (/10)	3.5 (3.4–3.7)	3.3 (3.2–3.5)	0.0022
**Total aHEI-2010 score (/100)**	**36.5 (36.2–36.8)**	**39.0 (38.5–39.4)**	**<0.** **0001**

**^a^** Adjusted for sex, age, income, education, food insecurity, marital status, region status, immigrant status, current smoking status, BMI, household size, and energy intake (continuous); **^b^** SSB stands for sugar-sweetened beverages; **^c^** EPA + DHA stands for eicosapentaenoic acid and docosahexaenoic acid; **^d^** PUFA stands for polyunsaturated fatty acid (excluding omega-3 fatty acids).

**Table 3 nutrients-15-00637-t003:** Adjusted **^a^** mean total alternate Healthy Eating Index 2010 (aHEI-2010) score and 95% confidence intervals, by characteristics of adults from the Canadian Community Health Survey (CCHS)-Nutrition in 2004 and 2015.

	Total aHEI-2010 Score		*p* Values	
	2004 Mean (95% CI)	2015 Mean (95% CI)	*p* for Time Effect	*p* for Interaction
**All**	36.5 (36.2–36.8)	39.0 (38.5–39.4)	<0.0001	
**Sex**				0.5793
Male	35.1 (34.6–35.6)	37.7 (37.1–38.4)	<0.0001	
Female	37.9 (37.4–38.3)	40.2 (39.6–40.8)	<0.0001	
*p* for sex effect	<0.0001	<0.0001		
**Age, years**				0.0019
19–30	33.0 (32.2–33.8)	35.5 (34.3–36.7)	0.0002	
31–50	36.0 (35.4–36.6)	39.3 (38.4–40.2)	<0.0001	
51–70	38.4 (37.8–39.0)	40.5 (39.8–41.3)	<0.0001	
≥71	38.9 (38.2–39.6)	39.4 (38.5–40.3)	0.2713	
*p* for age effect	<0.0001	<0.0001		
**Household income, quintiles**				0.0450
Q1	36.7 (35.8–37.5)	38.3 (37.2–39.4)	0.0121	
Q2	36.8 (36.0–37.7)	38.7 (37.7–39.7)	0.0059	
Q3	36.6 (35.9–37.4)	39.0 (38.2–39.9)	<0.0001	
Q4	35.7 (34.9–36.4)	39.2 (38.1–40.2)	<0.0001	
Q5	36.9 (36.2–37.7)	40.3 (39.1–41.4)	<0.0001	
*p* for income effect	0.7586	0.0128		
**Education**				0.0087
<High school	34.8 (34.1–35.5)	35.3 (34.0–36.7)	0.4864	
High school	34.9 (34.2–35.6)	37.1 (36.2–38.0)	0.0002	
Post-High school or certificate	35.6 (35.1–36.1)	38.5 (37.7–39.2)	<0.0001	
≥Bachelor’s degree	38.7 (37.9–39.4)	41.4 (40.6–42.1)	<0.0001	
*p* for education effect	<0.0001	<0.0001		
**Food insecurity**				0.1726
Yes	36.2 (35.0–37.5)	37.5 (36.1–38.8)	0.1752	
No	36.5 (36.2–36.8)	39.1 (38.6–39.6)	<0.0001	
*p* for food insecurity effect	0.6744	0.0304		
**Marital status**				0.1613
Married/Common-law	36.8 (36.3–37.2)	39.4 (38.7–40.0)	<0.0001	
Widowed/Separated/Divorced	36.6 (35.7–37.5)	38.0 (37.1–39.0)	0.0145	
Single/Never married	35.6 (34.9–36.4)	38.4 (37.4–39.5)	<0.0001	
*p* for marital status effect	0.0630	0.0640		
**Region status**				0.7740
Urban	36.6 (36.3–37.0)	39.2 (38.7–39.7)	<0.0001	
Rural	35.8 (35.2–36.4)	38.1 (37.0–39.2)	0.0002	
*p* for region status effect	0.0174	0.0688		
**Immigrant status**				0.5472
Yes	38.3 (37.6–38.9)	40.5 (39.6–41.4)	<0.0001	
No	35.9 (35.5–36.3)	38.5 (38.0–39.0)	<0.0001	
*p* for immigrant status effect	<0.0001	<0.0001		
**Current smoking status**				0.0214
Yes	33.4 (32.7–34.1)	34.5 (33.5–35.6)	0.0624	
No	37.1 (36.7–37.5)	39.9 (39.4–40.4)	<0.0001	
*p* for smoking status effect	<0.0001	<0.0001		
Body mass index, kg/m^2^				0.7026
<25.0	36.7 (36.2–37.3)	39.6 (38.9–40.3)	<0.0001	
25.0–29.9	36.9 (36.3–37.5)	39.1 (38.3–39.9)	<0.0001	
≥30.0	35.3 (34.5–36.0)	38.4 (37.4–39.4)	<0.0001	
*p* for BMI effect	0.0009	0.0448		

**^a^** Adjusted for sex, age, income, education, food insecurity, marital status, region status, immigrant status, current smoking status, BMI, household size, and energy intake (continuous), except for the stratification variable.

**Table 4 nutrients-15-00637-t004:** Adjusted **^a^** mean energy intake and alternate Healthy Eating Index 2010 (aHEI-2010) total score and 95% confidence intervals by province of residence of adults from the Canadian Community Health Survey (CCHS)-Nutrition in 2004 and 2015.

	Total energy intake (Kilocalories/day)			Total aHEI-2010 Score			
Provinces	2004	2015	*p*	2004	2015	Difference (95% CI)	*p*
**British Columbia**	2241 (2170–2312)	1821 (1766–1877)	<0.0001	38.1 (37.2–38.9)	42.0 (40.7–43.4)	4.0 (2.4, 5.5)	<0.0001
**Alberta**	2007 * (1938–2076)	1952 * (1871–2033)	0.3176	35.9 * (35.0–36.7)	39.6 * (38.5–40.7)	3.7 (2.4, 5.1)	<0.0001
Saskatchewan	2046 * (1962–2130)	1810 (1687–1932)	0.0015	35.6 * (34.7–36.5)	37.2 * (36.1–38.3)	1.7 (0.2, 3.1)	0.0240
Manitoba	2010 * (1939–2080)	1923 (1829–2017)	0.1391	35.0 * (34.1–35.8)	36.4 * (35.2–37.7)	1.5 (–0.1, 3.0)	0.0619
**Ontario**	1985 * (1945–2025)	1841 (1791–1890)	<0.0001	36.8 * (36.2–37.3)	39.2 * (38.5–40.0)	2.5 (1.5, 3.4)	<0.0001
**Quebec**	2189 (2120–2257)	2022 * (1967–2078)	0.0002	36.1 * (35.4–36.8)	37.7 * (36.7–38.6)	1.6 (0.4, 2.8)	0.0098
New Brunswick	1991 * (1910–2072)	1832 (1754–1910)	0.0072	35.5 * (34.4–36.6)	37.2 * (36.1–38.2)	1.7 (0.2, 3.2)	0.0252
Nova Scotia	2089 * (1991–2187)	1920 (1835–2006)	0.0109	36.1 * (34.9–37.4)	38.3 * (37.2–39.4)	2.2 (0.5, 3.9)	0.0111
Prince Edouard Island	1971 * (1887–2054)	1886 (1804–1969)	0.1462	35.3 * (34.1–36.4)	38.0 * (36.6–39.4)	2.7 (0.9, 4.5)	0.0033
Newfoundland and Labrador	1954 * (1853–2054)	1732 (1617–1847)	0.0037	33.1 * (32.1–34.1)	36.1 * (34.8–37.4)	3.0 (1.3, 4.6)	0.0004
*p* for province effect **	<0.0001	<0.0001		<0.0001	<0.0001	0.1854	

**^a^** Adjusted for age, sex, income, education, food insecurity, marital status, immigrant status, current smoking status, BMI, household size, and energy intake (continuous); * statistically significant different from British Columbia in the same year, *p* < 0.05. ** Time-by-province interaction *p* for total energy intake <0.0001.

**Table 5 nutrients-15-00637-t005:** Distribution of alternate Healthy Eating Index 2010 (aHEI-2010) scores in adults from the Canadian Community Health Survey (CCHS)-Nutrition in 2004 and 2015.

aHEI-2010, total score	0–24.99	25–49.99	50–74.99	75–100	<50
2004	20.2%	65.0%	14.5%	0.3%	**85.2%**
2015	14.6%	63.6%	21.4%	0.4%	**78.2%**
**aHEI-2010, item score**	**0–2.49**	**2.5–4.99**	**5.0–7.49**	**7.5–10**	**<5**
**Vegetables**					
2004	36.6%	25.0%	16.0%	22.4%	**61.6%**
2015	42.1%	25.8%	15.1%	17.0%	**67.9%**
**Fruits**					
2004	54.2%	22.2%	11.2%	12.4%	**76.4%**
2015	48.8%	25.3%	13.8%	12.1%	**74.1%**
**Whole grains**					
2004	67.4%	15.8%	7.4%	9.4%	**83.2%**
2015	64.8%	16.0%	8.2%	11.0%	**80.8%**
**SSB ^a^ and fruit juice**					
2004	51.9%	6.6%	2.0%	39.5%	**58.5%**
2015	45.8%	2.4%	1.9%	49.9%	**48.2%**
**Nuts and legumes**					
2004	74.3%	7.5%	3.5%	14.7%	**81.8%**
2015	67.5%	8.2%	4.7%	19.6%	**75.7%**
**Red/processed meat**					
2004	32.7%	11.4%	13.7%	42.2%	**44.1%**
2015	28.1%	9.7%	13.7%	48.5%	**37.8%**
**Long-chain n-3 (EPA + DHA) ^b^**					
2004	67.4%	16.0%	4.4%	12.1%	**83.5%**
2015	65.6%	16.1%	5.3%	13.0%	**81.7%**
**PUFA ^c^**					
2004	28.9%	34.1%	21.6%	15.4%	**63.0%**
2015	16.1%	29.7%	25.4%	28.8%	**45.8%**
**Sodium**					
2004	29.9%	19.8%	21.6%	28.7%	**49.7%**
2015	28.2%	18. 5%	24.0%	29.3%	**46.7%**
**Alcohol**					
2004	6.9%	75.0%	4.2%	14.0%	**81.8%**
2015	6.6%	77.6%	3.5%	12.3%	**84.2%**

**^a^** SSB stands for sugar-sweetened beverages; **^b^** EPA + DHA stands for eicosapentaenoic acid and docosahexaenoic acid; **^c^** PUFA stands for polyunsaturated fatty acid (excluding omega-3 fatty acids).

## Data Availability

The data are not publicly available due to their containing of information that could compromise the privacy of research participants.

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
