# Peer review of "Canadians Adults Fail Their Dietary Quality Examination Twice"

_nutrients, 2023, doi:10.3390/nu15030637_

Round 1

Reviewer 1 Report

Abstract Ok; keywordsOK; Introduction Ok; Materilas and MethodsOK, Discussion ; paragraph line 70 a 71 correspond more to a conclusions than a discussion"Teh overall picture of diet quality in Canadá indicates that more efforts and innovative strategies are neede to improve the situation". Conclussions; It is suggested to expand the conclussions.File attached

Author Response

Dear Sir, Madam,

We are so grateful for your significant input to improve our paper. You mainly urge us to expand the conclusions, including diet of immigrants, relationship of diet with income and educational level, validation of the instrument, as well a proposal of criteria that is what is discussed.

What we change ?

We re-worked the conclusion. We wrote:

"Dietary quality indices that include several validated health-related items are important tools for monitoring population health. To achieve this, they must be able to distinguish in their ratings between foods that have health benefits and those that do not. This does not seem to be the case for the C-HEI-2005 and might explain why it gives an overestimated score. For clinicians and policy makers, validated tools that adequately evaluate diet quality are useful in guiding toward the most appropriate approaches to improve the dietary scores of individuals or groups of individuals in our populations. Unfortunately, these multidimensional dietary tools are not widely used to monitor the status (deterioration or improvement) of overall nutrition. Neverthe-less, low dietary quality is one of the most important contributors to premature death in our societies. Given escalating food prices and socioeconomic disparities, lack of ac-tion on environmental and behavioural determinants, and food agency, poor diet quality in the Canadian population is likely to remain a concern for many years to come.

In conclusion, our results show that Canadians failed their diet quality test twice, both in 2004 and 2015. The use of the aHEI-2010 suggests a lower diet quality score among Canadians than that obtained with the C-HEI-2005, more comparable to the score among Americans. Behind the slight increase in the aHEI-2010 score in 2015 compared to 2004, it appears that not all individuals benefit from the same improve-ment in their food quality score, indicating disparities. In order to inform targeted policies and interventions, further analyses are required to address the disparity observed across provinces of Canada, to investigate the diet quality of immigrants and the rela-tionship between diet quality, income, and educational level. As a whole, significant efforts should be devoted to improving diet quality in Canadian adults."

Reviewer 2 Report

It's my pleasure to review this manuscript. This work is important for public health. For me, the manuscript is well written, especially for the Discussion, which discussed many good and interesting points.

Below are some small points for revision:

1.     The Introduction should expand more on assessing tools for diet quality. What are the differences between C-HEI-2005 and aHEI-2010? Why aHEI-2010 is better than C-HEI-2005? Is there any other tool/index assessing diet quality?

2.     Please define the abbreviation of SES in LINE54.

3.     Please use the abbreviation for body mass index in LINE98.

4.     The statement of Line 120-121 may be moved to the Introduction or the Discussion instead of the Conclusion.

Author Response

Dear Sir, Madam,

We are so grateful for your marked input to improve our manuscript. You requested to expand more the introduction on assessing tools for diet quality. We re-worked the manuscript and defined SES (socioeconomic status) in Line 54, using the abbreviation for body mass index in LINE98. We further moved the statement of Line 120-121 to the Discussion instead of the Conclusion, as you suggested. 

Reviewer 3 Report

This is a well written and interesting manuscript

Minor suggestions

The page/line numbering seems to have gone astray so hopefully it is clear where I am suggesting changes.

·         In section 2.2 it would be helpful to add the highest possible score for the aHEA-2010 I text- I appreciate is it shown in table 1 but as the tool was modified with removal of trans- fatty acid from scoring this would help with clarity

·         Discussion, p1, paragraph 1 lines 12-13= the phrasing of this sentence doesn’t make grammatical sense- rathe than “than ours among Canadians adults” please change to something like “These results are of a similar magnitude to those observed in our study of Canadian adults”

·         Discussion p1, line 33- please change the tense of this sentence to the past tense or update 2019 if a new CFG is planned

Author Response

Dear Sir, Madam,

We are so grateful for your marked input to improve our manuscript. 

In section 2.2 it would be helpful to add the highest possible score for the aHEA-2010 I text- I appreciate is it shown in table 1 but as the tool was modified with removal of trans- fatty acid from scoring this would help with clarity

Response: We re-worked the sentence, and wrote: "The remaining 10-items and scoring method of the aHEI-2010 (maximum score of 100) is described in the Table 1. "

Discussion, p1, paragraph 1 lines 12-13= the phrasing of this sentence doesn’t make grammatical sense- rathe than “than ours among Canadians adults” please change to something like “These results are of a similar magnitude to those observed in our study of Canadian adults”

Response: We re-phrased the sentence. We wrote: "These results are of a similar magnitude (36.5/100 in 2004 to 39.0/100 in 2015) to those observed in our study in Canadian adults."

Discussion p1, line 33- please change the tense of this sentence to the past tense or update 2019 if a new CFG is planned

Response: We re-phrased the sentence. We wrote: "In 2019, Health Canada released its new Canada's Food Guide (CFG-2019)19, which was a hybrid of the 2011 Harvard Healthy Plate20 and the 2015 Brazilian Food Guide21."